# The Activation of the Fibrodysplasia Ossificans Progressiva-Inducing ALK2-R206H Mutant Depends on the Distinct Homo-Oligomerization Patterns of ACVR2B and ACVR2A

**DOI:** 10.3390/cells13030221

**Published:** 2024-01-25

**Authors:** Szabina Szófia Szilágyi, Wiktor Burdzinski, Jerome Jatzlau, Marcelo Ehrlich, Petra Knaus, Yoav I. Henis

**Affiliations:** 1Department of Neurobiology, George S. Wise Faculty of Life Sciences, Tel Aviv University, Tel Aviv 6997801, Israel; sabinasi@tauex.tau.ac.il; 2Institute for Chemistry and Biochemistry, Freie Universität Berlin, 14195 Berlin, Germanyjerome.jatzlau@fu-berlin.de (J.J.); petra.knaus@fu-berlin.de (P.K.); 3Berlin-Brandenburg School for Regenerative Therapies (BSRT), Charité-Universitätsmedizin Berlin, 13353 Berlin, Germany; 4Shmunis School of Biomedicine and Cancer Research, George S. Wise Faculty of Life Sciences, Tel Aviv University, Tel Aviv 6997801, Israel; marceloe@tauex.tau.ac.il

**Keywords:** ALK2, ALK2-R206H, activin receptors, FOP (fibrodysplasia ossificans progressiva), activin A, FRAP, lateral diffusion, signaling

## Abstract

Mutations in activin-like kinase 2 (ALK2), e.g., ALK2-R206H, induce aberrant signaling to SMAD1/5/8, leading to Fibrodysplasia Ossificans Progressiva (FOP). In spite of extensive studies, the underlying mechanism is still unclear. Here, we quantified the homomeric and heteromeric interactions of ACVR2A, ACVR2B, ALK2-WT, and ALK2-R206H by combining IgG-mediated immobilization of one receptor with fluorescence recovery after photobleaching (FRAP) measurements on the lateral diffusion of a co-expressed receptor. ACVR2B formed stable homomeric complexes that were enhanced by Activin A (ActA), while ACVR2A required ActA for homodimerization. ALK2-WT, but not ALK2-R206H, exhibited homomeric complexes unaffected by ActA. ACVR2B formed ActA-enhanced heterocomplexes with ALK2-R206H or ALK2-WT, while ACVR2A interacted mainly with ALK2-WT. The extent of the homomeric complex formation of ACVR2A or ACVR2B was reflected in their ability to induce the oligomerization of ALK2-R206H and ALK2-WT. Thus, ACVR2B, which forms dimers without ligand, induced ActA-independent ALK2-R206H clustering but required ActA for enhancing the oligomerization of the largely dimeric ALK2-WT. In contrast, ACVR2A, which undergoes homodimerization in response to ActA, required ActA to induce ALK2-R206H oligomerization. To investigate whether these interactions are translated into signaling, we studied signaling by the FOP-inducing hyperactive ALK2-R206H mutant, with ALK2-WT signaling as control. The activation of SMAD1/5/8 signaling in cells expressing ALK2-R206H alone or together with ACVR2A or ACVR2B was measured by blotting for pSMAD1/5/8 and by transcriptional activation assays using BRE-Luc reporter. In line with the biophysical studies, ACVR2B activated ALK2-R206H without ligand, while activation by ACVR2A was weaker and required ActA. We propose that the homodimerization of ACVR2B or ACVR2A dictates their ability to recruit ALK2-R206H into higher complexes, enabling the homomeric interactions of ALK2-R206H receptors and, subsequently, their activation.

## 1. Introduction

The transforming growth factor-β superfamily (TGF-β-SF) includes 33 cytokines in humans. They are vital for biological processes throughout development, proliferation, differentiation, apoptosis, and more [1,2,3,4,5,6,7]. These cytokines play crucial roles in several diseases, including cancer [1,8,9], pulmonary arterial hypertension [10,11], and Fibrodysplasia Ossificans Progressiva (FOP), a rare, devastating genetic disease which causes abnormal bone formation, known as heterotopic ossification. FOP is caused by an activating mutation in the activin receptor-like kinase-2 (ALK2, encoded by the *ACVR1* gene), the most prevalent of which is the single nucleotide mutation where Arg206 is replaced by His (R206H) [12,13,14,15,16].

Crystallographic studies on complexes between the extracellular domains of ligand-bound TGF-β-SF receptors suggested heterotetrameric complexes between several type II and type I receptors [17,18,19,20,21,22]. Of note, the formation of type I or type II homodimers and their association into heterotetrameric complexes was also found for several full-length type I/type II TGF-β-SF receptors (the type I TGF-β receptor ALK5 and TβRII, and the bone morphogenetic protein (BMP) type I ALK3 or ALK6 with BMPRII) at the surface of live cells [23,24,25]. Following ligand binding to the cell-surface type I/type II receptor complexes, the type II receptor phosphorylates and activates the type I receptor, initiating SMAD-dependent or -independent signaling cascades [26,27,28,29]. The phosphorylated receptor-specific SMADs (SMAD2/3 or SMAD1/5/8) associate with the common SMAD (SMAD4) and translocate to the nucleus, where they activate or repress the transcription of multiple genes [30,31,32]. Depending on the receptors and ligands, there are two SMAD branches. TGF-β and activins induce mainly SMAD2/3 phosphorylation (via ALK5 for TGF-β and ALK4/7 for activins), while BMPs activate SMAD1/5/8 via ALK1/2/3/6 [32,33]. However, in some cases, TGF-β and activins may also signal to SMAD1/5/8 via ALK1 (for TGF-β) or ALK2 (for activins and TGF-β) [34,35,36,37,38,39], and BMPs may induce some signaling to Smad2/3 [40,41,42]. Moreover, ActA can form a non-signaling complex with ALK2 and type II receptors, portraying a crucial regulatory mechanism that dampens the aberrant ActA response in FOP [16,43,44].

ALK2 is of special interest since specific activating mutations, such as ALK2-R206H, result in abnormal activin signaling to SMAD1/5/8, inducing FOP [7,12,16,45]. ALK2-R206H has partial aberrant constitutive signaling to SMAD1/5/8, which is enhanced by activin A (ActA) in the presence of Act or BMP type II receptors [14,15,38,46,47,48,49]. However, in spite of extensive studies, the molecular mechanism underlying the aberrant signaling by ALK2-R206H is not fully clear. Recent studies [38,49] proposed that ActA mediates the clustering of ALK2 (WT or R206H). They suggested that unlike ALK2-WT, which is activated following phosphorylation by another type I receptor (ALK4 or ALK7), ALK2-R206H is activated by autophosphorylation following ActA-mediated clustering, which is induced by binding to a type II activin receptor (ACVR2B or ACVR2A). However, these studies left some open questions. Thus, one study measured the clustering of fluorescent-labeled ActA with no direct measure of receptor aggregation [38], while the other proposed a model where an unknown inhibitory “X factor” attaches to the cytoplasmic domain of ALK2-WT but not ALK2-R206H [49]. Moreover, neither the ability of ACVR2B and ACVR2A to form homomeric complexes nor the relation between their homodimerization and their ability to recruit ALK2-R206H into mutual oligomers were measured quantitatively. Given the proposed role for ALK2-R206H homomeric interactions in its aberrant signaling to SMAD1/5/8 in FOP, ACVR2A and ACVR2B may contribute differentially to ALK2-R206H activation. 

Here, we explored these questions by quantitative studies of (i) the percentage of homomeric and heteromeric complex formation between the above full-length receptors at the surface of live cells, (ii) their modulation by ActA, and (iii) the effects of these complexes on ALK2-R206H signaling to the SMAD1/5/8 pathway. The formation of receptor complexes was studied by patch/FRAP (fluorescence recovery after photobleaching of a receptor carrying a specific extracellular epitope tag and its modulation by crosslinking and immobilization of a co-expressed, differently tagged receptor). This method, which we developed and employed to study the interactions between multiple full-length TGF-β-SF receptors [23,24,25,50], allows us to measure the interactions between different receptors (heteromeric complexes) or between two versions of the same receptor carrying different epitope tags (homomeric complexes). Our studies demonstrated that ACVR2B forms stable homomeric complexes to a much higher degree than ACVR2A, which requires ActA for homodimerization. Unexpectedly, ALK2-R206H exhibited a much lower tendency to form homomeric complexes than ALK2-WT. Moreover, ACVR2B formed ActA-enhanced complexes with either ALK2-R206H or ALK2-WT, while ACVR2A interacted mainly with ALK2-WT. In line with the ability of ACVR2B to form homomeric complexes already without ligand, its co-expression with ALK2-R206H augmented the homodimerization of the latter prior to ActA stimulation. On the other hand, the induction of ALK2-R206H homodimerization by ACVR2A occurred only in the presence of ActA. Of note, aberrant signaling to SMAD1/5/8 by ALK2-R206H in U2OS cells supported these data and demonstrated a significantly stronger effect of ACVR2B, independent of its kinase activity. We propose a model where the degree of the homodimerization of the type II receptors (ACVR2B or ACVR2A/ActA) dictates their ability to recruit ALK2-R206H molecules into hetero-oligomeric receptor complexes, serving as hubs that enable homomeric interactions of and aberrant signaling by ALK2-R206H. 

## 2. Materials and Methods

### 2.1. Reagents

ActA (human, recombinant; cat. #120-14P) was from PeproTech (Rocky Hill, NJ, USA). Media and all other reagents for cell culture, such as antibiotics (penicillin–streptomycin) and Hanks’ balanced salt solution (HBSS), were purchased from Biological Industries Israel (Beit Haemek, Israel). Bovine serum albumin (BSA) without fatty acids (cat. #10-775-835-001) was obtained from Roche Diagnostics (Manheim, Germany). Protease inhibitor cocktail (cat. #P8340), 4-(2-hydroxyethyl)-1-piperazineethanesulfonic acid (HEPES), and Na_3_VO_4_ were from Sigma-Aldrich (St. Louis, MO, USA). Opti-MEM (cat. #11058021) was purchased from Gibco Life Technologies (Carlsbad, CA, USA). The QuikChange kit for site-directed mutagenesis (cat. # 200513) was from Agilent (Santa Clara, CA, USA).

### 2.2. Antibodies 

The 9E10 anti-myc tag monoclonal antibody (mouse αmyc; cat. #626802) [51] and the anti-HA (influenza hemagglutinin) epitope tag antibody HA.11 (rabbit αHA; cat. #902302) were obtained from BioLegend (San Diego, CA, USA). The αHA murine monoclonal antibody 12CA5 (cat. #11-66-606-001) was purchased from Roche Diagnostics. Where needed, 9E10 and 12CA5 IgGs were digested to Fab’ fragments as described [50]. Goat anti-rabbit (GαR) IgG labeled with Alexa Fluor 488 (Alexa 488-GαR; cat. #R37116), goat anti-mouse (GαM) F(ab’)_2_ labeled with Alexa 546 (cat. #A-11018), and Alexa 488-GαR F(ab’)_2_ (cat. #A-11070) were all obtained from Invitrogen-Molecular Probes (Eugene, OR). To convert the fluorescent F(ab′)_2_ to Fab′ fragments, they were reduced, and the free SH groups formed were blocked by iodoacetamide [50]. Rabbit antibodies to detect phospho (p) SMAD1/5/8 (cat. #13820) or total (t) SMAD1/5/8 (cat. #6944) were obtained from Santa Cruz Biotechnology (Santa Cruz, CA), and mouse anti-β-actin (cat. #0869100-CF) was obtained from MP Biomedicals (Solon, OH, USA). Normal goat γ-globulin (NGG; cat. #005-000-002), peroxidase-conjugated GαM (cat. #115-035-062), and GαR (cat. #115-035-144) IgGs were from Jackson ImmunoResearch Laboratories (West Grove, PA, USA).

### 2.3. Plasmids

Human ACVR2A in the pcDNA3.1 vector was obtained from Prof. G. Blobe (Duke University, Durham, NC, USA); we introduced myc or HA epitope tags at its N-terminus as described [50]. N terminally HA-tagged human ACVR2B (in pcDNA3.1) was generated by inserting the HA tag using overlapping PCR after nucleotide 66 [52]. ACVR2B (human) with N-terminal myc tag (cat. #LS-N12733) in pCMV3 was obtained from LSBio (Seattle, WA, USA). Kinase dead (KD) myc-ACVR2B-K217R mutant was generated from myc-ACVR2B using QuikChange site-directed mutagenesis kit. Human ALK2-WT with N-terminal HA tag in pCMV5 was described [50], and N terminally myc-tagged human ALK2-WT in pCMV3 (cat. #HG14875-NM) was obtained from Sino Biological (Wayne, PA, USA). The R206H point mutants of the epitope-tagged ALK2 were made using site-directed mutagenesis with the QuikChange kit. All constructs were verified by sequencing. The SMAD1/5/8-responsive luciferase reporter construct BRE-Luc in pGL3 [53] was obtained from Addgene (Watertown, MA, USA; cat. #45126). pRL-TK (Renilla Luciferase, cat. #E2810) was from Promega (Madison, WI, USA). 

### 2.4. Cell Culture 

Cell lines (COS7, (cat. #CRL-1651; U2OS, cat. #HTB-96)) were obtained from the American Type Culture Collection (ATCC, Manassas, VA, USA). The cells were grown in Dulbecco’s modified Eagle’s medium (DMEM) containing 10% FCS, antibiotics and L-glutamine at 37 °C, 5% CO_2_ [50]. The U2OS human cell line was authenticated by STR analysis at Microsynth AG (Balgach, Switzerland). All cells were routinely analyzed for potential contamination by mycoplasma using reverse transcriptase-PCR (RT-PCR) and found to be clean.

### 2.5. Transfection

Transfection of cells employed the TransIT-LT1 Mir2300 (cat. #MIR 2305, Mirus Bio, Madison, WI, USA) transfection reagent as per the protocol supplied by the manufacturer. Transfection of cells for Patch/FRAP studies was performed on cells growing in 6-well plates on glass coverslips, while for SMAD1/5/8 phosphorylation assays, they were grown directly in these plates. Transfection was with combinations of expression vectors for myc- or HA-tagged receptors. The transfection employed 0.3 to 1 μg DNA of each vector, selecting the amount such that the cell surface expression levels would be similar, determined by quantitative point confocal immunofluorescence as described [24,50]. DNA was complemented to a total of 1 μg with empty vector. For transcriptional activation assays using the Dual-Luciferase Reporter (DLR) assay, U2OS cells grown in 96-well plates were co-transfected with (i) 100 ng luciferase reporter construct (BRE-Luc); (ii) 30 ng pRL-TK (Renilla luciferase construct); (iii) 12.5 ng vectors encoding various receptors (see figure legends). Cells were subjected to the different assays 17–24 h after transfection, as described under each method and in the figure legends.

### 2.6. Fluorescent Labeling of Cells and IgG-Induced Crosslinking for FRAP and Patch/FRAP 

COS7 cells were transfected with expression vectors for the indicated myc- and/or HA-tagged receptors. After 24 h, they were serum-starved with 1% serum for 30 min and washed with cold buffer (HBSS supplemented with 20 mM HEPES pH 7.4 and 2% BSA; HBSS/HEPES/BSA. This was followed by blocking with 200 μg/mL (30 min, 4 °C) of NGG. For FRAP experiments, epitope-tagged receptors on the live cells were then labeled at 4 °C (thus labeling only the receptors localized at the cell surface) in the same buffer with murine Fab’ αmyc or αHA (40 μg/mL), followed by Alexa 546-Fab’ GαM (40 μg/mL). To conduct patch/FRAP experiments, the transfected cells were labeled, successively, as follows: (i) 40 μg/mL murine αmyc Fab’ and 20 μg/mL HA.11 rabbit αHA IgG; (ii) 40 μg/mL Alexa 546-Fab′ GαM together with 20 μg/mL Alexa 488-IgG GαR. This protocol leads to IgG-mediated crosslinking (CL) and immobilization of the HA-tagged receptor, while the myc-tagged receptor is not crosslinked and is labeled exclusively by monovalent Fab′. The lateral diffusion of the Fab’-labeled receptor is measured by FRAP. Where indicated in the figure legends, ligand (4 nM ActA) was added at the step of antibody labeling and kept in all incubations during the FRAP experiments.

### 2.7. FRAP and Patch/FRAP

Cells transfected with different combinations of receptors carrying HA or myc tags were labeled with fluorescent antibodies, as described in the preceding section. The labeled cells were taken for FRAP and patch/FRAP experiments as described by us previously [23,50]. A schematic representation of patch/FRAP experiments is depicted in Appendix A. To avoid internalization, the FRAP experiments were carried out at 15 °C for up to 20 min per sample. The beam of an Innova 70C (Coherent, Santa Clara, CA, USA) argon ion laser was focused by a Planapochromat 63x/1.4 NA oil-immersion objective via a fluorescence microscope (Axioimager.D1; Carl Zeiss MicroImaging, Jena, Germany) to a Gaussian spot of 0.77 ± 0.03 μm at the fluorescence at the spot illuminated by the laser beam was measured using monitoring intensity (528.7 nm, 1 μW), followed by a brief bleaching pulse of the same beam (5 mW, 20 ms), bleaching 60 to 75% of the fluorescence. Fluorescence recovery was followed by the monitoring beam. The *D* and *R_f_* values of each curve were derived by fitting to a lateral diffusion process using nonlinear regression [54]. Patch/FRAP studies were carried out similarly, except that a co-expressed epitope-tagged receptor was patched and immobilized by IgGs prior to the measurement [23,50]. Patch/FRAP measures the effects of immobilizing one receptor (e.g., HA-tagged) by IgG crosslinking on the lateral diffusion of another (myc-tagged, Fab’-labeled; see Appendix A). Of note, it is capable of detecting complex formation between either different receptors (heteromeric complexes) or two differently tagged variants of the same receptor (homomeric complexes). Moreover, it differentiates between transient and stable interactions [23,50].

### 2.8. SMAD1/5/8 Phosphorylation Measurements by Western Blot Analysis

U2OS cells were transfected, incubated for 24 h, and then starved for 2 h with 1% serum. They were stimulated (or not) for the periods mentioned in the figure legends by ActA (4 nM). Cells were lysed on ice (30 min) with lysis buffer (420 mM NaCl, 50 mM HEPES, 5 mM EDTA, 1% NP-40, 3 mM dithiothreitol, protease inhibitor cocktail, and 0.1 mM Na_3_VO_4_). Cell debris was removed by low-speed centrifugation, and the lysates were subjected to SDS-PAGE (10% polyacrylamide) followed by immunoblotting [50]. The blots were probed (12 h, 4 °C) by rabbit anti-pSMAD1/5/8 (1:1000), rabbit anti-tSMAD1/5/8 (1:1000), or murine anti-β-actin (1:50,000), followed by peroxidase-GαR or -GαM IgG (1:5000, 1 h). The bands were visualized by enhanced chemiluminescence (ECL) using Clarity ECL substrate (cat. #1705060, Bio-Rad, Hercules, CA, USA), recorded using ChemiDoc Touch imaging system (Bio-Rad), and quantified by Image Lab 5.0 software (Bio-Rad).

### 2.9. Transcriptional Activation Assays

U2OS cells grown in 96 wells were transfected as described under “transfection” with the BRE-Luc reporter plasmid and Renilla luciferase (pRL-TK), along with vectors encoding one or more of the receptors under study (HA-ALK2-WT, HA-ALK2-R206H, myc-ACVR2A, myc-ACVR2B, myc-ACVR2B-KD), as indicated in the figure legends. At 17 h post-transfection, the cells were serum-starved (5 h) and stimulated (or not) with 2 nM ActA for 19 h. They were then lysed in passive lysis buffer (cat. #E1910; Promega), and the luciferase activity was measured using a TECAN initiate f200 Luminometer (TECAN, Mannedorf, Switzerland). The results were normalized for transfection efficiency using the Renilla luminescence.

### 2.10. Statistical Analysis

Statistical significance was analyzed by Prism9 (GraphPad Software, San Diego, CA, USA) using one-way ANOVA followed by post hoc Bonferroni test or Student’s two-tailed *t*-test. Data are presented as mean ± SEM, along with the number of repetitions. A *p*-value below 0.05 was considered statistically significant.

### 2.11. Illustration

The graphical illustration shown in Figure 9 was created with CorelDRAW X8 and BioRender (https://biorender.com accessed on 18 December 2023).

## 3. Results

### 3.1. ACVR2B vs. ACVR2A and ALK2-WT vs. ALK2-R206H Display Different Tendencies to Form Homomeric Complexes

It was recently proposed [38,49] that aberrant activation of ALK2-R206H involves its ActA-mediated clustering. However, the potential contribution of ACVR2A vs. ACVR2B to the homomeric interactions of ALK2-R206H and the dependence of this contribution to the extent of the homodimerization of the ACVR2 receptors remained unclear. Therefore, we conducted quantitative studies on the ability of each receptor to form homomeric complexes, followed by studies on the ability of ACVR2A/B to affect the homomeric interactions of the ALK2 variants.

To measure the mode (stable vs. transient) and extent of interactions between receptor pairs at the surface of live cells, we employed patch/FRAP [23,24,50]. Using this method, we demonstrated the formation of stable or transient complexes on the FRAP time scale between TGF-β-SF receptors [23,24,50]. In patch/FRAP, one tagged receptor is patched and laterally immobilized by cross-linking with a double layer of IgGs; the effect on the lateral diffusion of a differently tagged co-expressed receptor (labeled by monovalent Fab’ fragments) is measured by FRAP. Complex formation between the receptors may reduce either *R_f_* or *D* of the Fab’-labeled receptor, depending on the FRAP time scale relative to the rates of the dissociation–association kinetics of the complex. Complex lifetimes longer than the characteristic FRAP times (i.e., stable interactions) reduce *R_f_* since bleached Fab’-labeled receptors do not have enough time to exchange between the immobilized complex and the surroundings. On the other hand, short complex lifetimes (transient interactions) enable each Fab’-labeled receptor to undergo several association/dissociation cycles during the FRAP measurement, reducing the effective *D* value without altering *R_f_* [23,24]. This method can also measure homomeric complexes, providing that two differently tagged versions of the same receptor are co-expressed. However, for homomeric complexes, a statistical correction is needed to reflect the probability of the formation of complexes containing similarly tagged receptors [23,24]. For homodimeric complexes between myc- and HA-tagged versions of the same receptor, the probabilities for homodimerization are 1:2:1 for myc/myc, myc/HA, and HA/HA-containing dimers, respectively. Moreover, in case the HA-tagged receptors are immobilized and the myc tags are labeled by Fab’ and measured by FRAP, the myc/myc dimers (which remain mobile) contain two myc tags per dimer, as opposed to one tag in mixed-tag (HA/myc) dimers. Therefore, myc/myc dimers are labeled by αmyc Fab’ at twice the intensity as myc/HA dimers. Thus, the % reduction measured in *R_f_* of the Fab’-labeled myc-tagged receptor is only half of the % homodimerization and should be multiplied by a factor of 2 [23,24].

We first conducted FRAP studies to measure the tendency of ACVR2B or ACVR2A to form homomeric complexes. These studies employed COS7 cells, which were used to characterize the lateral diffusion of multiple TGF-β-SF receptors [23,24,50] and, thus, provide a basis for comparison. To this end, we expressed myc-ACVR2B alone or together with HA-ACVR2B under conditions yielding similar cell-surface expression levels (see Materials and Methods); analogous studies were conducted on myc-ACVR2A alone or co-expressed with HA-ACVR2A. Where indicated, ActA (4 nM) was added; this saturating concentration was chosen based on our previously published studies [50]. Typical FRAP curves of myc-ACVR2B alone or co-expressed with HA-ACVR2B before and after IgG crosslinking of the latter are depicted in Figure 1A–D. The average mean ± SEM of multiple measurements under each condition are shown in Figure 1E,F. The *D* and *R_f_* values of singly expressed ACVR2B (4 × 10^−2^ μm^2^/s and 70%, respectively) were in the same range measured for other TGF-β-SF receptors [23,24,25,50]. The co-expression of myc-ACVR2B with HA-ACVR2B without crosslinking of the latter had no effect on the *R_f_* and *D* of myc-ACVR2B, in line with the two receptors being identical except for the N-terminal extracellular tag (compare the two leftmost bars in Figure 1E,F). Of note, the IgG-mediated crosslinking of HA-ACVR2B significantly reduced the *R_f_* of myc-ACVR2B without an effect on *D* (compare the two middle bars in Figure 1E,F). A reduction in *R_f_* alone characterizes stable interactions between the HA- and myc-tagged receptors on the FRAP timescale [23,50]. Thus, the *R_f_* of myc-ACVR2B was reduced from 70 to 53%, suggesting % homodimerization = 2 × [(70 − 53)/70] × 100 = 48%. In samples where HA-ACVR2B was immobilized by IgG crosslinking, ActA induced a significant further reduction in the *R_f_* (down to *R_f_* = 44%) of myc-ACVR2B (compare the two rightmost bars in Figure 1E,F); this suggests enhanced homomeric interactions upon ActA binding (% homodimerization = 2 × [(70 − 44)/70] × 100 = 74%). These effects are on the interactions between the receptors since, in control experiments, ActA had no effect on the lateral diffusion of singly expressed ACVR2B, as well as all the other receptors investigated in the current studies (ACVR2A, ALK2-WT, ALK2-R206H; Appendix A).

We then proceeded to measure the effects of immobilizing HA-ACVR2A by IgG crosslinking on the lateral diffusion of myc-ACVR2A. The lateral diffusion parameters of singly expressed myc-ACVR2A were close to those of myc-ACVR2B, with a somewhat higher *R_f_* (*D* = 3.8 × 10^−2^ μm^2^/s, *R_f_* = 82%; Figure 1G,H). Co-expression with HA-ACVR2A without IgG crosslinking had no effect on the lateral diffusion of myc-ACVR2A (as expected), and the immobilization of HA-ACVR2A had no significant effect on either *D* or *R_f_* of co-expressed myc-ACVR2A (compare the two middle bars in Figure 1G,H). This suggests that the homomeric interactions among ACVR2A molecules are below the detection limit of our method and much weaker than those of ACVR2B. Interestingly, homomeric complexes of ACVR2A were induced by ActA (% homodimerization = 2 × [(82 − 67)/82] × 100 = 37%), albeit to a lower level than for ACVR2B. It should be noted that the lack of effect of crosslinking HA-ACVR2A on myc-ACVR2A diffusion serves as a control to demonstrate the specificity of the immobilization effects measured. Additional controls [50,52] demonstrated that immobilizing an unrelated HA-tagged receptor (type II TGF-β receptor) has no effect on myc-ACVR2A and that immobilizing HA-ACVR2B does not alter the diffusion of another unrelated receptor (myc-neuropilin-1).

Next, we characterized the homomeric complex formation of ALK2-WT or ALK2-R206H. Singly expressed myc-ALK2-WT exhibited lateral diffusion in the typical range of transmembrane receptors (*D* = 4 × 10^−2^ μm^2^/s, *R_f_* = 70%; Figure 2A,B). As expected, mere co-expression with HA-ALK2-WT did not significantly alter the lateral diffusion parameters of myc-ALK2-WT. However, IgG-crosslinking of HA-ALK2-WT resulted in a mild but significant reduction in *R_f_* of myc-ALK2-WT (compare the two middle bars in Figure 2A,B). This suggests % homodimerization = 2 × [(70 − 62)/70] × 100 = 23%. ActA addition had no further effect on ALK2-WT homomeric interactions (compare the two rightmost bars in Figure 2A,B), in line with ActA binding much weaker to ALK2 as compared to ACVR2A/B, where it enhanced ACVR2A or 2B homomeric interactions. 

Unexpectedly, analogous studies on ALK2-R206H demonstrated a significantly weaker tendency to form homomeric complexes on its own. While the lateral diffusion parameters of singly expressed myc-ALK2-R206H were close to those of ALK2-WT (*D* = 4 × 10^−2^ μm^2^/s, *R_f_* = 62%; Figure 2C,D), they were not altered by co-expression with HA-ALK2-R206H, either without or with IgG crosslinking (compare the three leftmost bars in Figure 2C,D). As in the case of ALK2-WT, the addition of ActA had no significant effect on ALK2-R206H homomeric interactions (rightmost bars in Figure 2C,D).

### 3.2. Heterocomplex Formation of ACVR2A/B with ALK2 (WT or R206H) and the Effects on ALK2 Homomeric Interactions

It was recently proposed that the aberrant signaling by the ALK2-R206H mutant is induced by its homomeric clustering, which requires binding to ACVR2A/B and ActA [38,49]. Such a mechanism requires heterocomplex formation between ACVR2A/B and ALK2. To test the ability of the two ACVR2 variants to form heteromeric complexes with ALK2, we co-expressed myc-ACVR2A or myc-ACVR2B alone or together with HA-ALK2-WT or HA-ALK2-R206H. The effects of co-expression and IgG-mediated immobilization of the HA-tagged ALK2 variants on the lateral diffusion parameters of Fab’-labeled myc-tagged ACVR2B (Figure 3) or ACVR2A (Figure 4) were measured by patch/FRAP. As shown in Figure 3A (for myc-ACVR2B) and Figure 4A (for myc-ACVR2A), the cell-surface expression levels of the ACVR2 proteins were not altered by co-expression with ALK2 (WT or R206H). Furthermore, the surface expression levels of all the epitope-tagged receptors employed in the current studies (myc-ACVR2A, myc-ACVR2B, myc-ACVR2B-KD, HA-ALK2-WT, and HA-ALK2-R206H) in COS7 cells were in the same range (Appendix A). In the case of ACVR2B (Figure 3), the expression of HA-ALK2 (WT or R206H) without crosslinking had no effect on the *R_f_* or *D* of myc-ACVR2B (compare the two leftmost bars in Figure 3B,D). The IgG-crosslinking of HA-ALK2-WT or HA-ALK2-R206H markedly reduced the *R_f_* of myc-ACVR2B (from 70 to 48% and from 70 to 55%, respectively), leaving *D* unaltered (Figure 3B,D). Since, for heterocomplex formation, there is no need for statistical correction, this is indicative of stable complex formation between 31% of the ACVR2B population with ALK2-WT vs. 21% in the case of ACVR2B with ALK2-R206H. In both cases, ActA significantly enhanced the reduction in the *R_f_* of myc-ACVR2B, suggesting 51% and 36% heterocomplex formation with ALK2-WT and ALK2-R206H, respectively. Thus, ACVR2B forms significant levels of stable heterocomplexes with both ALK2 variants, which are enhanced by ActA.

Interestingly, a different pattern is observed for heterocomplex formation between ACVR2A and the ALK2 variants. The interactions of myc-ACVR2A with HA-ALK2-WT (Figure 4B,C) resembled those of myc-ACVR2B, except for a small reduction in the *R_f_* of myc-ACVR2A already upon co-expression with HA-ALK2-WT. The reduction in the *R_f_* of myc-ACVR2A upon crosslinking HA-ALK2-WT indicated that 27% of the ACVR2A population formed stable complexes with ALK2-WT, increasing to 42% in the presence of ActA. These results are in agreement with our former measurement of the interactions between ACVR2A and ALK2-WT [50]. However, the interactions of myc-ACVR2A with HA-ALK2-R206H (Figure 4D,E) were much weaker than those of ACVR2B. No significant reduction in the *R_f_* (or *D*) of myc-ACVR2A was detected upon co-expression with HA-ALK2-R206H, and only a small reduction in its *R_f_* was observed upon IgG crosslinking of the ALK2 mutant, indicating only 13% of the myc-ACVR2A in heterocomplexes. The addition of ActA enhanced the % of these heterocomplexes to ~23% (Figure 4D), a value still lower than for ACVR2B (36%; Figure 3D).

In light of the different abilities of ACVR2A vs. ACVR2B and ALK2-WT vs. ALK2-R206H to form homomeric complexes, it was important to explore whether binding of ALK2 (WT or R206H) to the ACVR2 receptors enhances ALK2 homomeric clustering and whether such an enhancement is related to the tendency of ACVR2B and ACVR2A to form homo-oligomers. To this end, we conducted patch/FRAP studies on the effects of overexpressing untagged ACVR2A or ACVR2B (which are, therefore, not crosslinked and not labeled by the anti-tag antibodies) on the homomeric interactions between myc- and HA-tagged ALK2-WT (Figure 5) or ALK2-R206H (Figure 6). First, we measured the effects on the homomeric interactions of ALK2-WT. As shown in Figure 5A, the cell-surface levels of myc-ALK2-WT (whose lateral diffusion is measured in the current experiment) were unaffected upon co-expression with untagged ACVR2A or ACVR2B. To measure the effect of ACVR2A or ACVR2B on the homomeric interactions of ALK2-WT, cells were transfected with myc-ALK2-WT together with HA-ALK2-WT, without or with untagged ACVR2A, ACVR2B, or ACVR2B-KD. HA-ALK2-WT was immobilized (or not; control) by IgG crosslinking, and the effects on the lateral diffusion of myc-ALK2-WT were measured by FRAP in the absence or presence of ActA (Figure 5B,C). Comparison with the homomeric interactions of ALK2-WT in the absence of ACVR2A/B (taken from Figure 2A,B and shown on the left panels to enable direct comparison) showed that the overexpression of untagged, free (uncrosslinked) ACVR2A canceled the reduction in the *R_f_* of myc-ALK2-WT upon crosslinking of co-expressed HA-ALK2-WT (Figure 5B, middle panel). This is in line with our demonstration that ACVR2A does not form homodimers without ActA (Figure 1G) and gains further support by the finding (Figure 5B, middle panel) that ActA re-established the homomeric interactions of ALK2-WT, in accord with its ability to induce homomeric complexes of ACVR2A (Figure 1G). 

Analogous studies on the effects of overexpressing untagged ACVR2B, which forms homodimers already in the absence of ActA (Figure 1E), yielded different results. The reduction in the *R_f_* of myc-ALK2-WT upon immobilization of HA-ALK2-WT was unaffected by the overexpression of untagged ACVR2B but showed a significant further reduction upon addition of ActA (Figure 5B). These results are in accord with our demonstration that ActA enhances ACVR2B homomeric interactions (Figure 1E) and stabilizes the interactions between ACVR2B and ALK2-WT (Figure 3B). Of note, the overexpression of untagged ACVR2B-KD yielded similar results (Figure 5B,C, rightmost panels), demonstrating that the effects are independent of the kinase activity of the type II receptor.

We then conducted analogous studies on the effect of ACVR2A or ACVR2B on the homomeric interactions of ALK2-R206H (Figure 6). The experiments were as described in Figure 5, except that the cells were transfected with myc-ALK2-R206H and HA-ALK2-R206H in place of the ALK2-WT counterparts. Here, co-expressed myc-ALK2-R206H and HA-ALK2-R206H had no detectable homomeric interactions (left panels of Figure 6B,C; taken from Figure 2C,D and shown here to enable direct comparison). The overexpression of untagged (and, therefore, uncrosslinked) ACVR2A had no effect (no significant reduction in the *R_f_* of myc-ALK2-R206H in the presence of immobilized HA-ALK2-R206H), while addition of ActA induced a significant reduction in the *R_f_* of myc-ALK2-R206H upon immobilization of HA-ALK2-R206H (with no effect on *D*) under these conditions, suggesting the formation of a low level of homomeric ALK2-R206H complexes. These findings are in accord with the ability of ActA to enhance ACVR2A/ALK2-R206H interactions (Figure 4D) and to induce some homodimerization of ACVR2A (Figure 1G,H), which is then able to recruit monomeric ALK2-R206H molecules into mutual heteromeric complexes. Unlike ACVR2A, the overexpression of untagged ACVR2B together with myc-ALK2-R206H and IgG-crosslinked HA-ALK2-R206H (Figure 6B) induced a significant reduction in the *R_f_* of myc-ALK2-R206H (from 62 to 48%, in line with % homodimerization of 2 × [(62 − 48)/62] × 100 = 45%) already in the absence of ligands. The addition of ActA yielded similar results, most likely due to this high level of homodimerization being close to saturation. These findings are in accord with the significant level of homodimerization of ACVR2B already in the absence of ActA (Figure 1E) and with its ability to recruit ALK2-R206H into heteromeric complexes (Figure 3D). Analogous results were obtained upon overexpression of untagged ACVR2B-KD, indicating that the kinase activity of the type II receptor is not required for the measured interactions (Figure 6B).

### 3.3. ACVR2B Is More Effective than ACVR2A in Inducing Constitutive and ActA-Mediated Activation of ALK2-R206H Signaling to SMAD1/5/8

Our patch/FRAP experiments demonstrated major differences between the homomeric interactions of ALK2-WT and ALK2-R206H (Figure 2), as well as in the ability of ACVR2B vs. ACVR2A to induce or enhance the homomeric clustering of the ALK2 variants (Figure 5 and Figure 6). To investigate whether the formation of homomeric ALK2-R206H complexes correlates with aberrant signaling by ActA to SMAD1/5/8, we conducted signaling studies in U2OS osteosarcoma cells, which respond well to ActA signaling [50]. In these experiments, we focused on signaling to SMAD1/5/8 by the ALK2-R206H mutant since this mutant is proposed to be activated by homomeric clustering, which is not sufficient for the activation of ALK2-WT that requires phosphorylation by ALK4 or ALK7 [38,49]. Indeed, under our experimental conditions in U2OS cells, the expression of ALK2-WT alone or together with ACVR2B-KD, whose lack of activity was verified by the SMAD1/5/8-sensitive BRE-Luc transcriptional activation assay (Appendix A), did not promote significant signaling to SMAD1/5/8 (Figure 7D and Figure 8). First, to determine the optimal ActA stimulation time of the ALK2-R206H mutant in U2OS cells, the cells were transfected with HA-ALK2-R206H (or empty vector); 24 h post-transfection, they were starved (2 h, 37 °C) and stimulated with 4 nM ActA, the concentration determined earlier by us as most effective in U2OS cells; see ref. [50]. Signaling to SMAD1/5/8 was measured by immunoblotting for pSMAD1/5/8, tSMAD1/5/8, and β-actin (Appendix A). The strongest stimulation was obtained at 30 and 60 min; we selected 60 min stimulation for further experiments. Of note, the epitope-tagged HA-ALK2-R206H mutant has partial constitutive activity, which is enhanced by ActA (Appendix A), as reported earlier for the R206H mutant [14,38,46,47,48,49]. The activity of the other epitope-tagged receptor constructs used for the signaling studies in U2OS cells was validated as shown in Appendix A (for myc-ACVR2A and myc-ACVR2B), or measured by us recently HA-ALK2-WT; ref. [50]. Moreover, we showed [50] that HA-tagged and untagged ALK2-WT in U2OS cells respond similarly to ActA stimulation. The epitope-tagged receptors are also capable of binding ActA, as demonstrated by the ability of ActA to enhance homomeric and heteromeric interactions between the epitope-tagged receptors (Figure 1, Figure 3 and Figure 4). Although we cannot exclude some differences between tagged and untagged receptors, it appears that the major signaling features are retained in the tagged receptors.

Next, we studied the effect of co-expressing ACVR2A or ACVR2B on the ability of ALK2-R206H (or ALK2-WT as control) to signal to SMAD1/5/8. U2OS cells were transfected with HA-ALK2-R206H (or empty vector) alone or together with myc-tagged ACVR2A, ACVR2B, or ACVR2B-KD. At 24 h post-transfection, the cells were starved, stimulated (or not; control) with ActA, and subjected to lysis and Western blotting (Figure 7). Under these conditions, the cell-surface levels of the tagged receptors expressed in the signaling studies were similar, as determined by the fluorescence intensity of the Fab’-labeled receptors using the FRAP apparatus under non-bleaching conditions (Appendix A), as described in Figure 3A [24,50]. While the expression of ALK2-R206H (but not ALK2-WT; Figure 7B,D) induced some constitutive signaling to pSMAD1/5/8, the co-expression of ALK2-R206H with myc-ACVR2A in the absence of ligand did not enhance this signaling further (Figure 7A,C), in line with the finding that both ALK2-R206H and ACVR2A are mainly monomeric without ActA (Figure 1G and Figure 2C). On the other hand, the co-expression with ACVR2B, which forms homodimers already without ligand (Figure 1E), significantly elevated the constitutive signaling of ALK2-R206H to SMAD1/5/8 (Figure 7A,C). ActA induced some increase in pSMAD1/5/8 in cells co-expressing ACVR2A or ACVR2B. This is in line with the ActA-mediated enhancement of the homodimerization of ACVR2A (Figure 1G) or ACVR2B (Figure 1E) and with their increased ability to recruit ALK2-R206H to heteromeric complexes in the presence of ActA (Figure 3D and Figure 4D). Of note, ACVR2B-KD was as effective as ACVR2B-WT in the signaling induced by ALK2-R206H (Figure 7A,C) but not by ALK2-WT (Figure 7B,D), demonstrating that the kinase activity of the type II receptor is dispensable for its ability to stimulate ALK2-R206H signaling to SMAD1/5/8. 

To complement these studies, we examined whether the differential effects of ACVR2B and ACVR2A on the homomeric interactions of ALK2-R206H are also reflected in transcriptional activation of the SMAD1/5/8 pathway. To this end, we employed Dual-Luciferase Reporter gene assays using the SMAD1/5/8-responsive luciferase reporter construct BRE-Luc [53] as described under Materials and Methods. We compared the transcriptional activation of the SMAD1/5/8 pathway in U2OS cells transfected with HA-ALK2-R206H (or HA-ALK2-WT as control) alone or together with myc-tagged ACVR2A, ACVR2B, or ACVR2B-KD. Transfection with HA-ALK2-R206H alone slightly elevated the transcriptional activation of BRE-Luc, albeit below the significance limit, as also observed in the pSMAD1/5/8 signaling assay (Figure 7A,C). The addition of ActA significantly increased BRE-Luc activation in cells expressing ALK2-R206H (Figure 8), similar to pSMAD1/5/8 formation but to a higher degree (Figure 7C). Control experiments with U2OS cells expressing ALK2-WT did not show partial constitutive activation or ActA-mediated pSMAD1/5/8 formation under the conditions employed (Figure 8), in line with the aberrant signaling to SMAD1/5/8 being a property of the R206H mutant. The co-transfection of myc-ACVR2A with HA-ALK2-R206H did not have a significant effect on the basal BRE-Luc activity (in the absence of ActA), in line with pSMAD1/5/8 formation (Figure 7C), as well as with the homomeric nature of ACVR2A (Figure 1G) and its inability to induce homomeric clustering of ALK2-R206H (Figure 6B). The transcriptional activation response upon the addition of ActA was mildly enhanced in cells co-transfected with myc-ACVR2A and HA-ALK2-R206H, in accordance with the ability of ActA to induce ACVR2A homodimerization (Figure 1G). On the other hand, the co-transfection of myc-ACVR2B, which forms homodimers on its own (Figure 1E), with HA-ALK2-R206H led to a strong ligand-independent activation of BRE-Luc, which was further elevated by ActA (Figure 8). Importantly, the co-expression of myc-ACVR2B-KD with HA-ALK2-R206H (but not HA-ALK2-WT) led to similar results asco-expression with myc-ACVR2B (Figure 8). This suggests that the kinase activity of ACVR2B is not required for its effects on the ALK2-R206H-mediated transcriptional activation of the SMAD1/5/8 pathway.

## 4. Discussion

The molecular mechanisms that bridge between the FOP-causing mutations in ALK2 (e.g., R206H) and the excessive SMAD1/5/8 signaling that promotes aberrant ossification are still contentious and not fully characterized. It was proposed that the FOP mutations disrupt the binding of the inhibitory immunophilin FKBP12 to the glycine-serine rich (GS) domain of ALK2, leading to moderate constitutive activation of signaling to SMAD1/5/8 [13,46,55] and inducing FOP. However, later studies indicated that this is unlikely the sole mechanism and the constitutive activation could also be promoted by allosteric changes in the kinase domain [55,56]. Moreover, it was shown that the aberrant signaling by ALK2-R206H requires a type II receptor and is enhanced by ActA [15,16,38,49,56]. Recently, it was proposed that ActA, along with ACVR2A/B, mediate the homomeric clustering of ALK2-R206H, which enables its activation by autophosphorylation [38]. However, the latter study did not directly measure the homomeric clustering of ALK2-R206H or its dependence on the assembly of ACVR2A and/or ACVR2B into homodimers. Here, we investigated these issues by biophysical studies on the homomeric and heteromeric interactions between ACVR2A, ACVR2B, ALK2-R206H, and ALK2-WT; their modulation by ActA; and the effects on signaling to SMAD1/5/8 by ALK2-R206H. 

The formation of homomeric ALK2-R206H complexes is a crucial factor in the latter models. Therefore, we directly investigated the formation of homomeric complexes of each of the receptors involved. To this end, we took advantage of the ability of the patch/FRAP method to detect the formation and dynamics of homomeric complexes between two differently tagged versions of the same receptor at the plasma membrane of live cells [23,24]. First, we characterized the tendency of ACVR2B and ACVR2A to form homomeric complexes, following the effects of the IgG-mediated immobilization of HA-ACVR2B (or 2A) on the lateral diffusion of co-expressed myc-ACVR2B (or 2A) (Figure 1). As explained by us formerly [23,50] and under Results, reduction in *R_f_* of the Fab’-labeled myc-tagged receptor following lateral immobilization of its co-expressed HA-tagged counterpart demonstrates the formation of stable complexes between the differently tagged receptors, while reduction in the *D* value is indicative of transient interactions between the receptor pairs. These studies demonstrated a marked reduction in the *R_f_* of myc-ACVR2B (with no effect on *D*) following immobilization of HA-ACVR2B, which became more pronounced in the presence of ActA (Figure 1E). As explained under Results, for homodimerization, a statistical correction (multiplying the % reduction in *R_f_* by 2) is required to derive the % homodimerization [23,24]; thus, for ACVR2B, the % homodimerization was 48%, going up to 74% in the presence of ActA. This result is in line with the report that ActA enhanced the homodimerization of ACVR2B measured by a complementation-based nanoluciferase reporter assay [49]; however, the nanoluciferase-based assay yields only a relative measure of the increase in complexes between the cytoplasmic domains of the interacting receptors upon ligand stimulation and does not measure the basal level of the complexes. In the current studies, similar patch/FRAP experiments on the homomeric interactions of HA-ACVR2A with myc-ACVR2A demonstrated a significant level of homomeric complexes only in the presence of ActA (37%; Figure 1G). Thus, ACVR2B forms homomeric complexes in the absence of ligand, which are enhanced by ActA, while ACVR2A forms homomeric complexes to a lower degree only in the presence of ActA. These differences between the two ACVR2 receptors are highly relevant to their abilities to recruit ALK2-R206H into higher oligomers (see model in Figure 9).

Analogous experiments compared the homomeric interactions of ALK2-R206H with those of ALK2-WT. Of note, ALK2-WT exhibited a significantly higher level of stable homomeric complex formation (deduced from the reduction in *R_f_*, but not in *D*, of the Fab’-myc-ALK2-WT upon IgG crosslinking of HA-ALK2-WT) as compared to ALK2-R206H (Figure 2A,C). Here, 23% of the ALK2-WT population resided in homodimers, with no further effect of ActA. On the other hand, the *R_f_* or *D* values of myc-ALK2-R206H were not significantly reduced by the IgG-crosslinking of HA-ALK2-R206H, either with or without ActA, indicating that the mutant ALK2 does not form homomeric complexes to a measurable level. The lack of effect of ActA on homomeric interactions of the ALK2 variants is supported by the failure to detect effects of ActA on such complexes by the luciferase complementation assay [49] and is in accord with the weak binding of ActA to type I receptors [22,57,58,59].

While ALK2-WT displays a higher tendency than ALK2-R206H to form homomeric complexes on its own, ALK2-WT homo-oligomerization does not lead to activation (see Figure 7D), as it is not hyperactive (unlike ALK2-R206H) and requires phosphorylation by ALK4 or ALK7 to be activated [38]. On the other hand, ALK2-R206H, which has a mild constitutive activity, is thought to be further activated by autophosphorylation following its homomeric clustering. Therefore, such clustering would be expected to be mediated by the recruitment of monomeric ALK2-R206H molecules to heteromeric complexes with ACVR2B/A. To this end, we first tested the ability of ACVR2B and ACVR2A to recruit ALK2 (WT or R206H) into heteromeric complexes. Patch/FRAP studies indicated that stable heteromeric complexes are formed between ACVR2B and either ALK2-R206H or ALK2-WT (Figure 3B,D; 21% vs. 31% heterocomplex formation, respectively). ActA significantly increased the formation of these heterocomplexes to 36% and 51%, respectively. On the other hand, while heterocomplex formation between ACVR2A and ALK2-WT (Figure 4B; 27% in heterocomplexes without ligand, increasing to 42% with ActA) was similar to that observed for ACVR2B/ALK2-WT (Figure 3B; 31%, increasing to ~50% with ActA), the interactions between ACVR2A and ALK2-R206H were much weaker in the absence of ActA (Figure 4D; 13% in heterocomplexes, as compared to 21% with ACVR2B), increasing significantly in the presence of ligand (to 23%). We conclude that ACVR2B is more effective than ACVR2A in recruiting ALK2-R206H into heterocomplexes, and ActA enhances heteromeric complex formation in both cases. These differences could be related to the significant level of homodimers formed by ACVR2B relative to ACVR2A, resulting in higher avidity towards ALK2. This conclusion is supported by the elevated binding of ActA to the surface of COS7 cells expressing ACVR2B over ACVR2A [59].

A critical aspect of autophosphorylation-based mechanisms of ALK2-R206H activation is the induction of its homo-oligomerization by binding to type II receptors serving as scaffolds. To obtain a direct measure of the ability of ACVR2A or ACVR2B to affect the homomeric interactions of ALK2-WT (Figure 5) and ALK2-R206H (Figure 6), we designed patch/FRAP studies where the ability of co-expressed, untagged ACVR2A or ACVR2B on the degree of homomeric complex formation among differently tagged ALK2 variants was measured. The overexpression of untagged ACVR2A (which is mainly monomeric) led to a loss of the interactions between myc- and HA-tagged ALK2-WT, as indicated by the increase in the *R_f_* of myc-ALK2-WT back to the original level observed in the absence of HA-ALK2-WT crosslinking (Figure 5B). On the other hand, untagged ACVR2A had no effect on the homomeric interactions of ALK2-R206H (Figure 6B), which is monomeric to begin with (Figure 2C). The addition of ActA, which induces ACVR2A homodimerization (Figure 1G) and enhances ACVR2A/ALK2-WT and ACVR2A/ALK2-R206H heterocomplexes (Figure 4B,D), led to the reformation of ALK2-WT homomeric complexes (Figure 5B; 23% homodimerization) and induced such complexes of ALK2-R206H (Figure 6B; 25%). A plausible explanation for these results is that overexpressed ACVR2A, which is mainly monomeric without ligand, can bind only one ALK2-WT molecule (either myc- or HA-tagged). Under such conditions, this heterocomplex formation would be at the expense of homomeric ALK2-WT complexes. As for ALK2-R206H, since both ACVR2A and ALK2-R206H are monomeric in the absence of ligand, homomeric complexes of ALK2-R206H are not expected to form. In the presence of ligand, ActA-bound ACRVR2A undergoes homodimerization and, thus, can recruit two ALK2-WT or ALK2-R206H molecules, inducing homomeric interactions between them, this time within a heteromeric complex.

Unlike ACVR2A, ACVR2B formed homodimers already prior to ligand binding. Accordingly, in the absence of ligand, ACVR2B did not induce the dissociation of the homodimeric complexes of ALK2-WT (Figure 5B); rather, it caused highly significant homodimerization of the mainly monomeric ALK2-R206H (Figure 6B; 45% homodimerization). ActA, which enhances the homodimerization of ACVR2B and the heteromeric interactions of ACVR2B with either ALK2-WT or ALK2-R206H (Figure 3B,D), augmented ALK2-WT homodimerization (to 62%), with no further effect on ALK2-R206H homomeric interactions, which appear to be already saturated at the 45% level induced by ACVR2B. Of note, the level of homomeric clustering is important in the context of the ALK2-R206H mutant, as it is required for its activation but not for ALK2-WT signaling. To further establish the role of the type II receptor in ALK2-R206H signaling (see Figure 7 and Figure 8), we tested whether untagged ACVR2B-KD induces homomeric complexes of ALK2 (WT or R206H). As shown (Figure 5 and Figure 6), ACVR2B-KD induced ALK2-R206H and ALK2-WT homomeric interactions similar to ACVR2B, suggesting that the kinase activity of the type II receptor is not required [15,38]. 

Taken together, the biophysical studies suggest that ACVR2B forms homomeric complexes to a much higher degree than ACVR2A, which forms such complexes only in the presence of ActA (see model in Figure 9A). ALK2-R206H has a lower tendency than ALK2-WT to form homomeric complexes and such complexes are induced to a higher degree by ACVR2B (most likely due to its largely dimeric nature) than by the monomeric ACVR2A, which disperses ALK2-WT homodimers. Accordingly, in the absence of ActA, ACVR2B, but not ACVR2A, can induce homomeric complexes of ALK2-R206H by the recruitment of ALK2-R206H monomers to heteromeric complexes. On the other hand, ACVR2A requires ActA for a similar effect. These results are in line with the recent report that ACVR2B is more potent than ACVR2A in inducing the ActA-dependent homodimerization of ALK2-R206H and ALK2-WT [49].

The differences between the abilities of ACVR2B and ACVR2A to induce or enhance the homomeric clustering of ALK2-R206H and ALK2-WT give rise to the hypothesis that homomeric complexes of the type II receptors may serve as hubs for the recruitment of monomeric ALK2-R206H receptors into mutual complexes, resulting in aberrant signaling to SMAD1/5/8 in response to ActA. To explore whether the above differences in complex formation are translated to signaling, we investigated the effects of overexpressing ACVR2A/B on ALK2-R206H signaling to SMAD1/5/8 (and on ALK2-WT signaling as control) by Western blotting for pSMAD1/5/8 (Figure 7) and by the transcriptional activation of a SMAD1/5/8-responsive luciferase construct (BRE-Luc; Figure 8). These experiments employed U2OS cells, where we recently characterized signaling to SMAD1/5/8 by BMP9 and signaling to SMAD2/3 by ActA [50]. Combining signaling studies with siRNA knockdown of specific receptors, these studies demonstrated that BMP9 activates SMAD1/5/8 in these cells via the type II receptors ACVR2A and ACVR2B and the type I receptors ALK2 and ALK3. On the other hand, ActA induced signaling to SMAD2/3 by ACVR2A and ACVR2B, with ALK4 as the main type I receptor. Of note, in U2OS cells, ActA did not induce detectable signaling to SMAD1/5/8. Therefore, we now focus on the ActA-mediated signaling to SMAD1/5/8 upon the expression of the ALK2-R206H mutant in U2OS cells. In both pSMAD1/5/8 formation and BRE-Luc activation assays, singly expressed ALK2-R206H (but not ALK2-WT) exhibited a mild level of constitutive signaling, which can be due to a combination of several factors, such as diminished inhibition by FKBP12 and allosteric alterations in the kinase domain [55,56]. This basal signaling was enhanced by ActA in both assays, albeit to a higher degree in the luciferase assay. This may reflect the different experimental conditions (much longer incubation time and a different biological readout in the transcriptional activation assay) and/or the fact that the luciferase-based studies measure only the transfected cell population, while the entire cell population contributes to the Western blotting assay. 

In line with the notion that homodimerization of the type II receptors determines their effect on ALK2-R206H signaling, ACVR2B expression without ligand significantly enhanced ALK2-R206H constitutive signaling, while ACVR2A had no significant effect (Figure 7 and Figure 8). The lack of effect of ACVR2A is in accord with its monomeric nature (Figure 1G) and its inability to recruit monomeric ALK2-R206H (Figure 2C) into higher complexes (Figure 6B). The pronounced ability of ACVR2B to enhance the constitutive signaling of ALK2-R206H parallels its high degree of homodimerization in the absence of ligand (Figure 1E). The co-expression of ACVR2A or ACVR2B with ALK2-R206H led to a higher level of ActA-mediated signaling (Figure 7C and Figure 8), in line with the increase in ACVR2A (Figure 1G) or ACVR2B (Figure 1E) homodimerization in the presence of ActA. Of note, the recruitment of ALK2-R206H into heterocomplexes with ACVR2A or ACVR2B was also increased by ActA (Figure 3D and Figure 4D). The effects of the type II receptors on signaling to either pSMAD1/5/8 or BRE-Luc transcriptional activation were independent of the kinase activity of the type II receptor, as demonstrated by the similar ability of kinase-dead ACVR2B (ACVR2B-KD) to induce constitutive and ActA-mediated signaling by ALK2-R206H, but not by ALK2-WT (Figure 7D and Figure 8). The latter finding is in accord with former reports that ALK2-R206H activation does not depend on phosphorylation by the type II receptor [15,38]. 

Based on the current findings and previous reports [15,38,49,56], we propose an updated model for the activation mechanism of the aberrant signaling by ALK2-R206H to the SMAD1/5/8 pathway (Figure 9). The importance of such a model is highlighted by its relevance to the proposed pathological role of the R206H mutation in the etiology of FOP disease [12,13,14,15,16]. This mutation leads to the overactivation of the SMAD1/5/8 pathway and, most significantly, to its aberrant activation by ActA [14,15,38,46,47,48,49]. This pathway is critical for bone development and regeneration, and its overactivation in response to ActA leads to abnormal bone formation in soft tissues in FOP patients, severely restricting their movement [48,60]. In our proposed model, the homodimerization of the type II receptor (ACVR2B or ACVR2A) plays a crucial role, as they serve as oligomerization hubs that recruit ALK2-R206H molecules into mutual clusters, thus enabling their autophosphorylation. It should be noted that while ALK2-WT can also be recruited into such complexes, this does not result in aberrant signaling, as ALK2-WT is not constitutively active, and its activation requires the kinase activity of the type II receptor to activate another type I receptor (ALK4, ALK7), which in turn activates ALK2-WT [38]. As shown in Figure 9A, singly expressed ACVR2A and ALK2-R206H are mainly monomeric, while ACVR2B and ALK2-WT form a significant amount of homomeric complexes. ActA enhances the homomeric interactions of ACVR2B and induces ACVR2A homomeric complexes but has no effect on ALK2-R206H or ALK2-WT homomeric interactions, in line with its relatively low affinity to these type I receptors [22,57,58]. These properties determine the effect of the type II receptor on the recruitment of ALK2-R206H into mutual complexes where two ALK2-R206H molecules are brought into close proximity, thus enabling their autophosphorylation (Figure 9B). Here, ACVR2B can induce aberrant signaling by ALK2-R206H already without ligand due to its ability to recruit ALK2-R206H to multimeric clusters, enabling homomeric interactions of ALK2-R206H receptors and their activation. On the other hand, ACVR2A requires ActA to induce similar (albeit weaker) effects on ALK2-R206H due to its mainly monomeric state in the absence of ActA. This mechanism may have important implications for FOP and other ALK2-related diseases, such as diffuse intrinsic pontine glioma [12,13,14,15,16,38,61,62], and may serve to identify potential therapeutic targets. For example, the inability of the mainly monomeric ACVR2A to activate ALK2-R206H in the absence of ligand raises the possibility that an ACVR2A analog that does not bind ActA may be able to compete with other type II receptors for binding ALK2-R206H, thus reducing its activation. 

## Figures and Tables

**Figure 1 cells-13-00221-f001:**
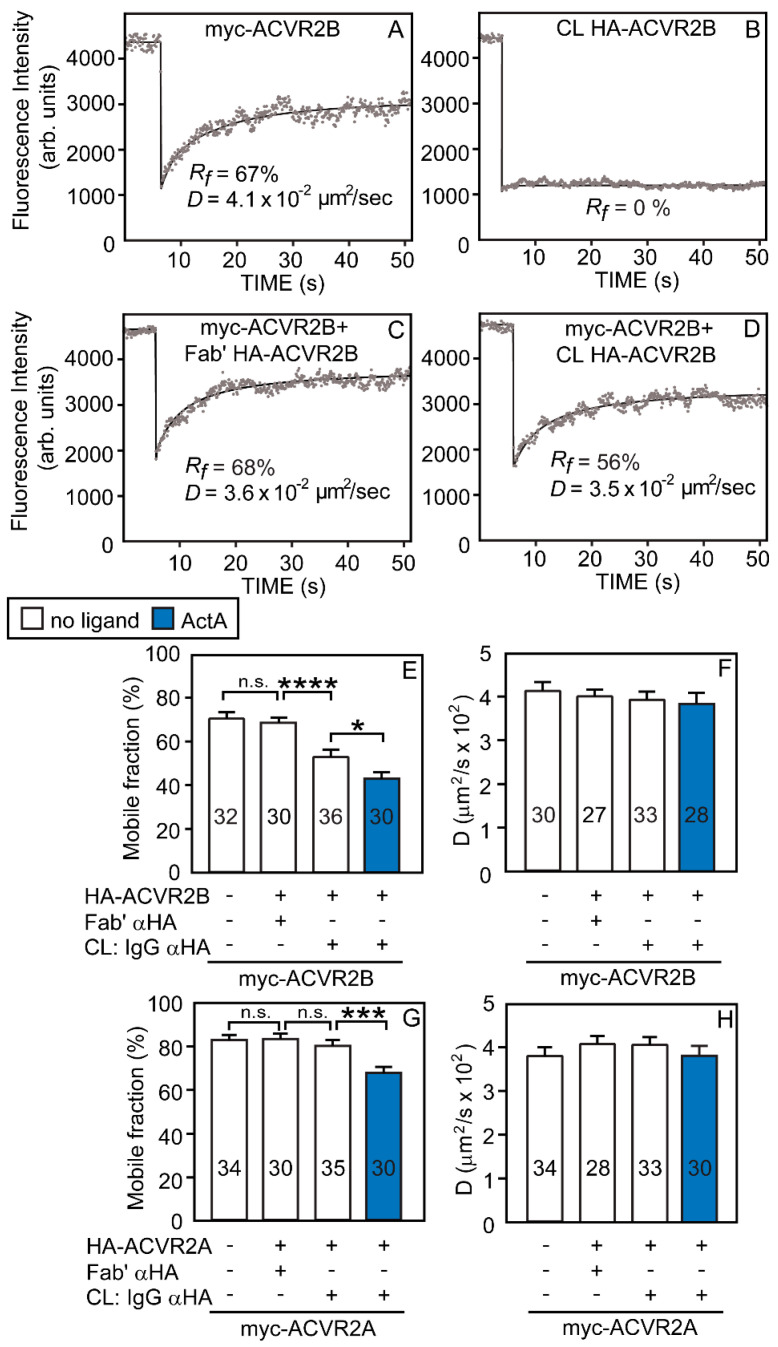
ACVR2B demonstrates a higher tendency than ACVR2A to form homomeric complexes. COS7 cells were transfected with expression vectors encoding mycACVR2A or -ACVR2B alone or together with the HA-tagged version of the same receptor (or empty vector; control). After 24 h, live cells were subjected to fluorescent antibody labeling and IgG-crosslinking (CL), as described under Materials and Methods. This protocol follows the scheme shown in Appendix A, where the HA-tagged receptor is patched/immobilized and labeled by Alexa 488-GαR IgG (designated CL: IgG αHA), and the myc-tagged version of the same receptor is labeled exclusively by monovalent Fab’ (with Alexa 546-GαM Fab’ as secondary antibody). In control experiments without crosslinking, the HA-tagged receptor was labeled by Fab’ instead of IgGs. Where indicated, ligand (4 nM ActA) was added at the last fluorescent labeling step and retained at later steps. FRAP studies were conducted at 15 °C to minimize internalization (see Materials and Methods). (**A**–**D**) Representative FRAP curves of singly-expressed myc-ACVR2B (**A**), IgG-crosslinked HA-ACVR2B (**B**), myc-ACVR2B co-expressed with uncrosslinked (Fab’-labeled), or IgG-crosslinked HA-ACVR2B ((**C**,**D**), respectively). Solid lines represent the best-fit (by non-linear regression) to the lateral diffusion equation (see Materials and Methods). The *R_f_* and *D* values of each representative FRAP curve are included within each panel. (**E**,**G**) Average *R_f_* values; (**F**,**H**) average *D* values. The bars depict the mean ± SEM values; the number of measurements (each conducted on a different cell) is shown within each bar. Asterisks indicate significant differences between the *R_f_* values of the pairs indicated by brackets (*, *p* < 0.01; ***, *p* < 2 × 10^−4^; ****, *p* < 10^−4^; n.s. = not significant; one-way ANOVA and Bonferroni post hoc test). The results suggest that a significant percentage of ACVR2B homomeric complexes forms already in the absence of ligands and is further enhanced by ActA (**E**). On the other hand, a significant level of homomeric complexes for ACVR2A is observed only in the presence of ActA (**G**). No significant differences were found between the *D* values (**F**,**H**).

**Figure 2 cells-13-00221-f002:**
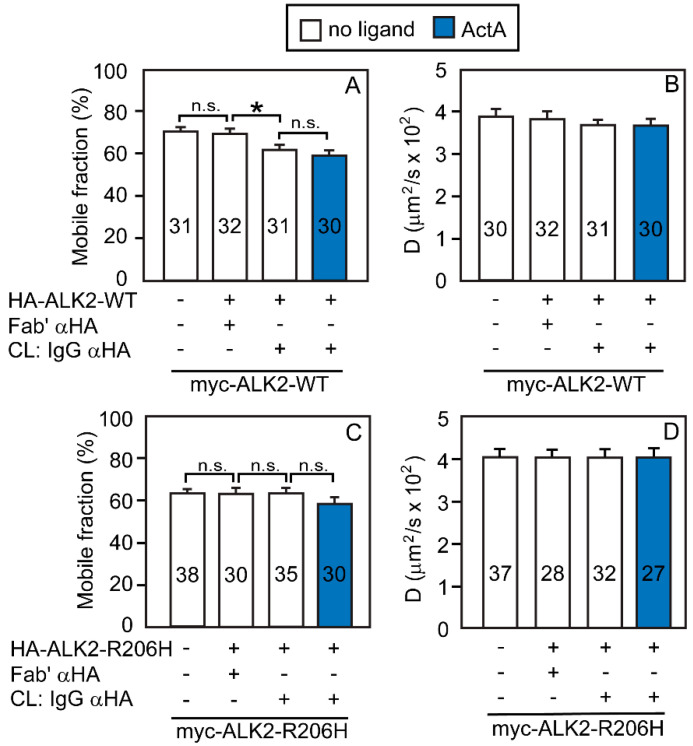
ALK2-WT, but not ALK2-R206H, forms a significant amount of homomeric complexes. COS7 cells were co-transfected with expression vectors encoding myc-tagged ALK2-WT or ALK2-R206H alone or together with their HA-tagged counterpart (or empty vector). Where shown, the HA-tagged receptor was immobilized by IgG-crosslinking as in Figure 1, following the schematic description in Appendix A. Thus, the HA-tagged receptor was patched and crosslinked by IgGs, and the myc-tagged version of the same receptor was labeled by Fab’ fragments. The lateral mobility of the Fab’-labeled myc-tagged receptor was measured by FRAP. Where indicated, ActA (4 nM) was added where indicated as in Figure 1. (**A**,**C**) Average *R_f_* values; (**B**,**D**) average *D* values. Bars are mean ± SEM; the number of measurements (each conducted on a different cell) is shown within each bar. Asterisks indicate significant differences between the *R_f_* values of the pairs indicated by brackets (*, *p* < 0.03; one-way ANOVA and Bonferroni post hoc test. n.s. = not significant). A similar analysis of the *D* values showed no significant differences.

**Figure 3 cells-13-00221-f003:**
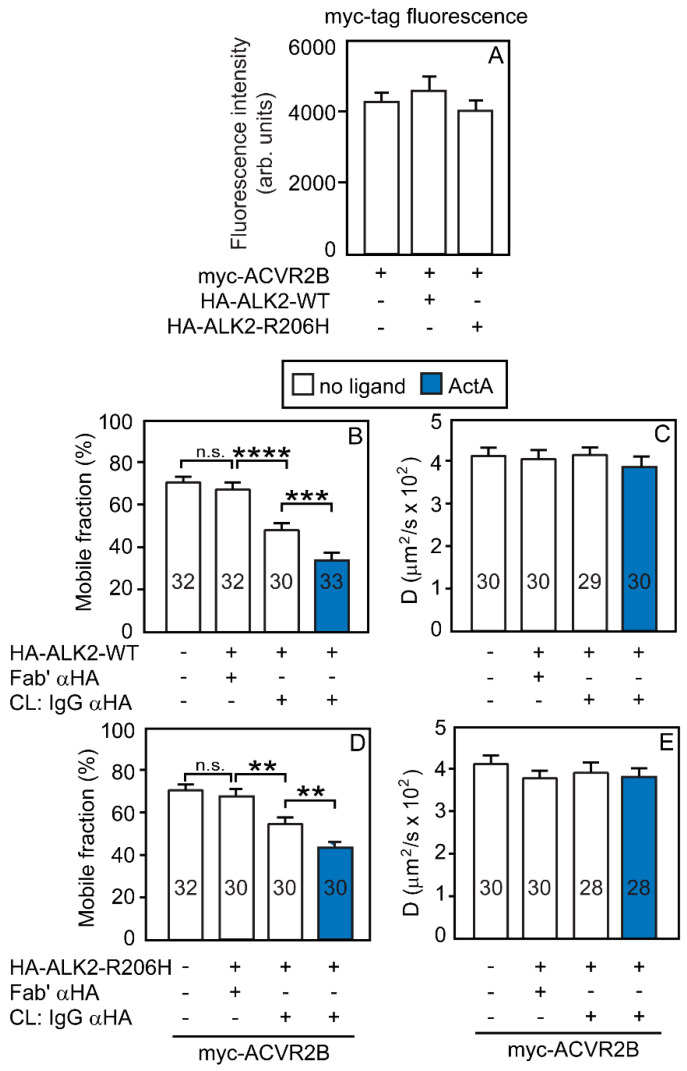
ACVR2B forms stable heteromeric complexes with either ALK2-WT or ALK2-R206H. COS7 cells were co-transfected with expression vectors encoding myc-ACVR2B alone or together with HA-tagged ALK2 (WT or R206H). Where indicated, HA-ALK2 was immobilized by IgG-crosslinking as in Figure 1, as shown schematically in Appendix A. The HA-tagged receptor (ALK2-WT or ALK2-R206H) was patched and crosslinked by IgGs, and the co-expressed myc-ACVR2B was labeled exclusively by monovalent Fab’; the lateral mobility of Fab’-labeled myc-ACVR2B was measured by FRAP, without or with ActA (4 nM). (**A**) Quantification of the cell surface levels of myc-ACVR2B alone or co-expressed with HA-tagged ALK2 variants. Myc-ACVR2B cell surface receptors were labeled at 4 °C by a saturating concentration (40 μg/mL) of murine Fab’ αmyc, followed by 40 μg/mL Alexa 546-Fab’ GαM, and fixed (4% paraformaldehyde). This protocol enables the measurement of the levels of the tagged receptors at the plasma membrane under identical conditions (same laser excitation line and intensity, same microscope filters, same settings of the photomultiplier tube) [24,50]. The surface levels of the receptors were quantified by measuring the fluorescence intensity from a point-confocal spot by the FRAP apparatus under non-bleaching conditions as described) [24,50]. Data are mean ± SEM of 30 measurements under each condition. No significant differences were observed between the values under the different conditions. (**B**,**D**) Average *R_f_* values; (**C**,**E**) average *D* values. Bars depict the average values (mean ± SEM); the number of measurements (each conducted on a different cell) is shown on each bar. Asterisks indicate significant differences between the *R_f_* values of the pairs indicated by brackets (**, *p* < 4 × 10^−3^; ***, *p* < 7 × 10^−4^; ****, *p* < 10^−4^; one-way ANOVA and Bonferroni post hoc test. n.s. = not significant). Similar analysis of the *D* values showed no significant differences.

**Figure 4 cells-13-00221-f004:**
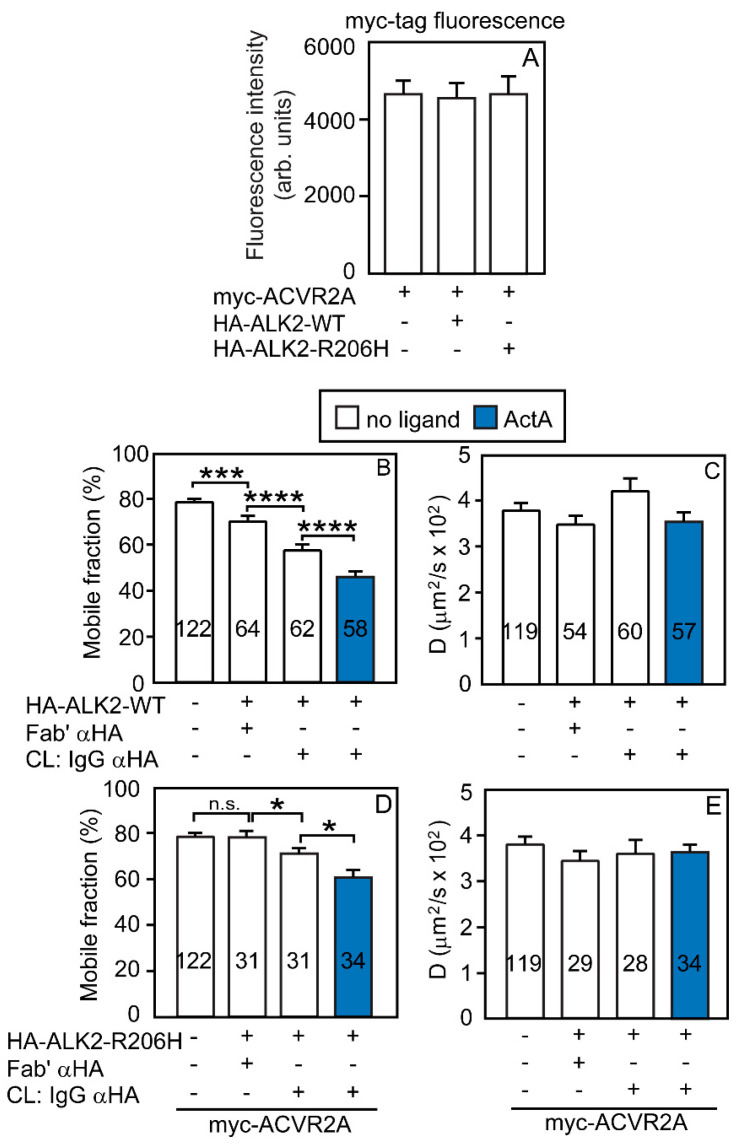
ACVR2A forms heteromeric complexes with ALK2-WT to a much higher degree than with ALK2-R206H. Experiments were as in Figure 3, following the scheme depicted in Appendix A, except that Fab’-labeled myc-ACVR2A replaced myc-ACVR2B. Where indicated, HA-ALK2-WT or HA-ALK2-R206H were immobilized by IgG-crosslinking, as in Figure 1. The lateral mobility of Fab’-labeled myc-ACVR2A was measured by FRAP, without or with ActA (4 nM; see Figure 1). (**A**) Quantification of the cell surface levels of myc-ACVR2A alone or co-expressed with HA-tagged ALK2 variants. The experiment was conducted exactly as described in Figure 3A. Data are mean ± SEM of 30 measurements under each condition. No significant differences were observed between the surface levels of myc-ACVR2A alone or co-expressed with HA-tagged ALK2 (WT or R206H). (**B**,**D**) Average *R_f_* values; (**C**,**E**) average *D* values. Bars depict the average values (mean ± SEM); the number of measurements (each conducted on a different cell) is shown on each bar. Asterisks indicate significant differences between the *R_f_* values of the pairs indicated by brackets (*, *p* < 0.03; ***, *p* < 8 × 10^−4^; ****, *p* < 10^−4^; one-way ANOVA and Bonferroni post hoc test. n.s. = not significant). Similar analysis of the *D* values showed no significant differences.

**Figure 5 cells-13-00221-f005:**
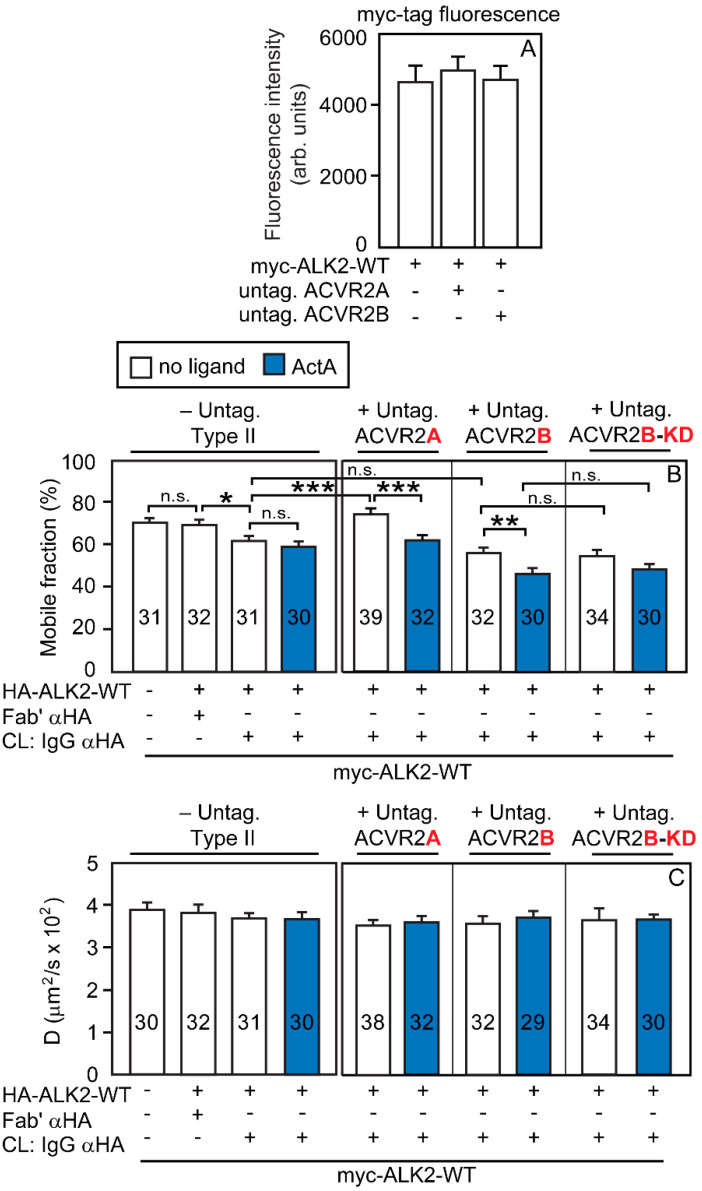
ACVR2B, but not ACVR2A, enhances ALK2-WT homomeric interactions in the presence of ActA. Patch/FRAP studies were conducted on COS7 cells expressing myc-ALK2-WT alone or together with HA-ALK2-WT. Where indicated, untagged ACVR2A, ACVR2B, or ACVR2B-KD were co-expressed. CL marks IgG-mediated crosslinking of HA-ALK2-WT, performed as in Figure 2A,B. The schematics of the experimental design are depicted in Appendix A; HA-ALK2-WT was patched and crosslinked by IgGs, while myc-ALK2-WT was labeled by Fab’ fragments. Where indicated, an untagged ACVR2 receptor variant was co-expressed as a third receptor to measure its effect. The lateral mobility of the Fab’-labeled myc-ALK2-WT was measured by FRAP. (**A**) Control experiments showing that the cell surface levels of myc-ALK2-WT are not significantly affected by co-expression with untagged ACVR2A or ACVR2B. The experiment was conducted as described in Figure 3A, except that the surface levels of myc-ALK2-WT were measured by the point confocal method (see Figure 3A). Data are mean ± SEM of 30 measurements in each case. No significant differences were observed between the surface levels of myc-ALK2-WT alone or co-expressed with untagged ACVR2A or ACVR2B (one-way ANOVA and Bonferroni post hoc test). (**B**) Average *R_f_* values; (**C**) average *D* values. Bars depict the average values (mean ± SEM); the number of measurements is depicted within each bar. Asterisks indicate significant differences between the *R_f_* values of the pairs indicated by brackets (*, *p* < 0.03; **, *p* < 9 × 10^−3^; ***, *p* < 2 × 10^−4^; one-way ANOVA and Bonferroni post hoc test. n.s. = not significant). Similar analysis of the *D* values showed no significant differences. The left panels in (**B**,**C**), designated “-Untag. Type II”, depict the *R_f_* and *D* values of myc-ALK2-WT co-expressed with HA-ALK2-WT without or with IgG αHA crosslinking; these values were taken from Figure 2A,B and are shown to enable direct comparison with the effects of co-expressing untagged ACVR2A/B. As shown, co-expression with untagged ACVR2A interfered with the formation of homomeric ALK2-WT complexes, which were restored in the presence of ActA (which dimerizes ACVR2A; Figure 1G). On the other hand, ACVR2B (which forms homodimers also without ActA; Figure 1E) did not disrupt ALK2-WT homomeric interactions and elevated them in the presence of ActA. Of note, ACVR2B-KD had the same effects as ACVR2B, demonstrating that the kinase activity of the type II receptor is not required.

**Figure 6 cells-13-00221-f006:**
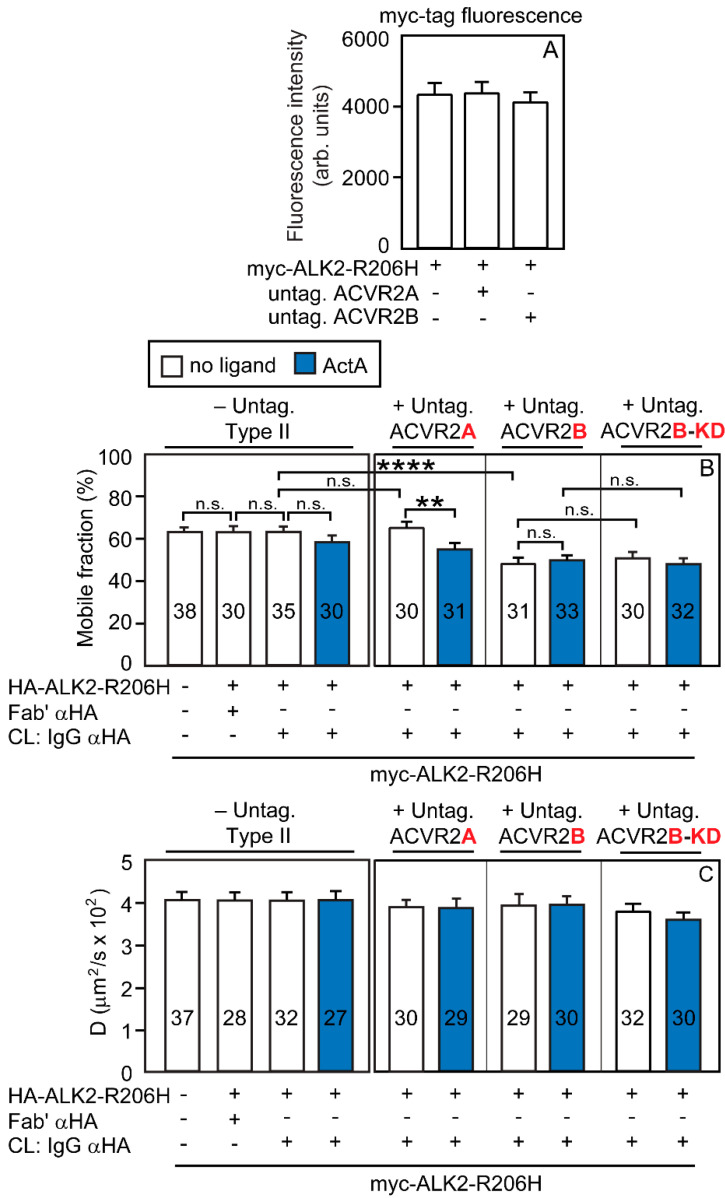
ACVR2B is more effective than ACVR2A in mediating ALK2-R206H homomeric interactions. Patch/FRAP studies were conducted exactly as described in Figure 5, except that myc-and HA-tagged ALK2-R206H replaced the tagged ALK2-WT constructs. The experimental scheme follows the one depicted in Appendix A, as explained in Figure 5. The lateral mobility of the Fab’-labeled myc-ALK2-R206H was measured by FRAP. (**A**) The cell surface levels of myc-ALK2-R206H are not altered by co-expression with untagged ACVR2A or ACVR2B. The experiment was conducted as described in Figure 3A, except that the surface levels of myc-ALK2-R206H were measured by the point confocal method (see Figure 3A). Data are mean ± SEM of 30 measurements in each case. No significant differences were observed between the surface levels of myc-ALK2-R206H expressed alone or together with untagged ACVR2A or ACVR2B (one-way ANOVA and Bonferroni post hoc test). (**B**) Average *R_f_* values; (**C**) average *D* values. Bars depict the average values (mean ± SEM); the number of measurements is depicted within each bar. Asterisks indicate significant differences between the *R_f_* values of the pairs indicated by brackets (**, *p* < 8 × 10^−3^; ****, *p* < 10^−4^; one-way ANOVA and Bonferroni post hoc test. n.s. = not significant). Similar analysis of the *D* values showed no significant differences. The left panels in (**B**,**C**), designated “-Untag. Type II”, depict the *R_f_* and *D* values of myc-ALK2-R206H co-expressed with HA-ALK2-R206H without or with IgG αHA crosslinking; these values were taken from Figure 2C,D and are shown to enable direct comparison with the effects of co-expressing untagged ACVR2A/B. Co-expression with untagged ACVR2A had no effect on ALK2-R206H homomeric interactions, which remained undetectable but were induced in the presence of ActA, conditions under which ACVR2A undergoes dimerization (Figure 1G). On the other hand, ACVR2B (which forms homodimers also without ActA; Figure 1E) induced ALK2-R206H homomeric interactions already without ActA. The effects of kinase-dead untagged ACVR2B-KD were indistinguishable from those of ACVR2B.

**Figure 7 cells-13-00221-f007:**
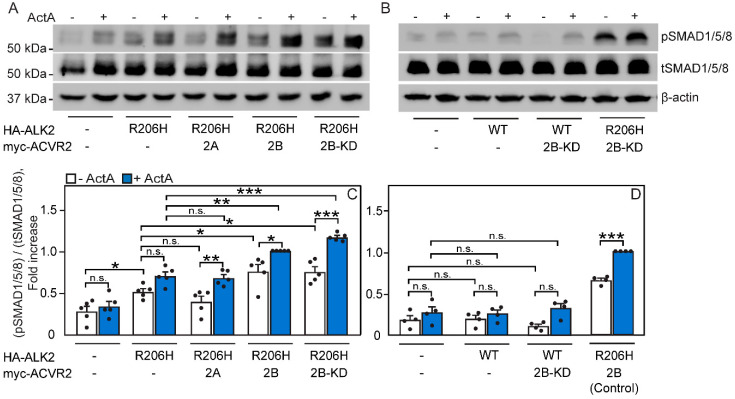
ALK2-R206H-mediated pSMAD1/5/8 formation is induced by ACVR2B more efficiently than by ACVR2A. U2OS cells were transfected with vectors encoding HA-ALK2-R206H (or HA-ALK2-WT) alone or with myc-tagged ACVR2A, ACVR2B, or ACVR2B-KD. After 24 h, cells were starved (2 h, 1% serum) and stimulated (or not; control) with ActA (4 nM, 60 min, 37 °C). Cells were lysed, subjected to SDS--PAGE, and immunoblotted for pSMAD1/5/8, tSMAD1/5/8, and β-actin. As shown in Appendix A, the cell-surface levels of the tagged receptors were similar and were not affected by the co-expressed receptors. (**A**,**B**) Representative blots of ActA signaling to pSMAD1/5/8. (**C**,**D**) Quantification of ActA-mediated pSMAD1/5/8 formation. The bands were visualized by ECL and quantified by densitometry. Data are mean ± SEM of the pSMAD1/5/8 over tSMAD1/5/8 ratio of 5 (**C**) or 4 (**D**) independent experiments. The value obtained for ActA-stimulated cells co-transfected with HA-ALK2-R206H and myc-ACVR2B was taken as 1. Asterisks show significant differences between the pairs indicated by brackets, using one-way ANOVA and Bonferroni post hoc test (*, *p* < 0.02; **, *p* < 4 × 10^−3^; ***, *p* < 8 × 10^−4^; n.s. = not significant).

**Figure 8 cells-13-00221-f008:**
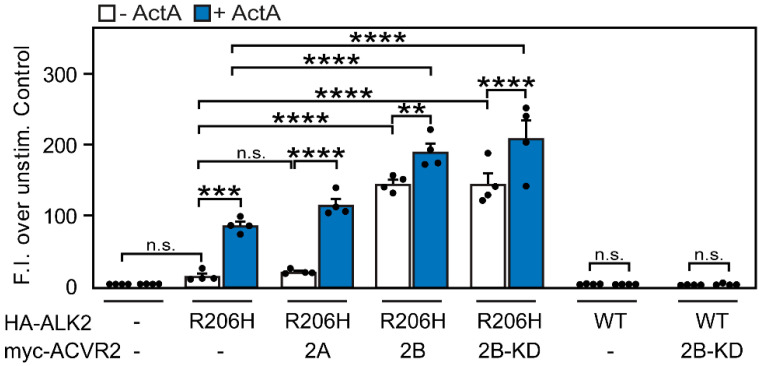
ACVR2B is superior to ACVR2A in eliciting ALK2-R206H-mediated transcriptional activation of the SMAD1/5/8 pathway. U2OS cells were co-transfected with BRE-Luc and pRL-TK, together with HA-ALK2-R206H or HA-ALK2-WT (alone or together with myc-ACVR2A, myc-ACVR2B, or myc-ACVR2B-KD). These constructs were replaced by empty vector for control samples. After 17 h, cells were starved without serum (5 h) and stimulated (or not; control) with ActA (2 nM, 19 h). Relative Luminescence Units (RLU) are expressed as mean fold induction  ±  SEM (n  =  4 independent experiments). The results were normalized for transfection efficiency using Renilla luminescence by the DLR luminescence assay. The value in untreated, unstimulated cells was taken as 1. The cell-surface levels of the tagged receptors were not altered by the co-expressed receptors (Appendix A). Asterisks show significant differences between the pairs indicated by the brackets, using one-way ANOVA and Bonferroni post hoc test (**, *p* < 1 × 10^−3^; ***, *p* < 5 × 10^−4^; ****, *p* < 10^−4^; n.s. = not significant).

**Figure 9 cells-13-00221-f009:**
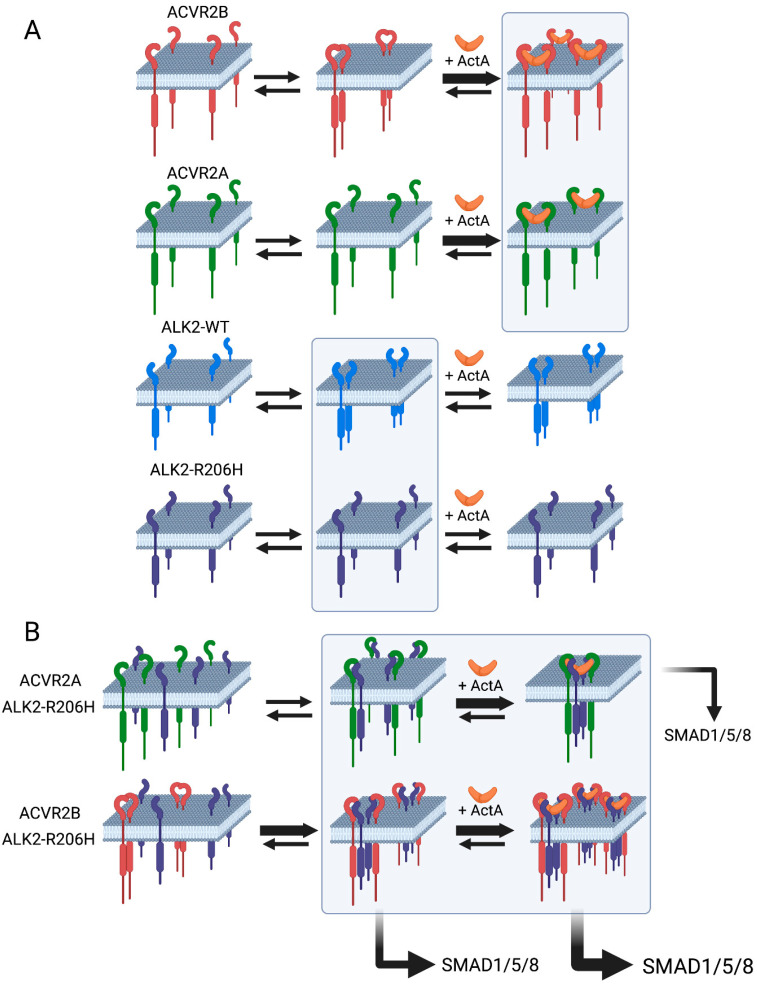
Model for the recruitment of ALK2-R206H into homomeric clusters by ACVR2A/B and its dependence on the dimeric nature of the type II receptors. (**A**) Homodimerization state of the singly-expressed receptors and the dependence on ActA. ACVR2B (red) forms stable homodimers already without ActA, which are enhanced by the ligand (thicker black arrow). ACVR2A (green) requires ActA (orange) to form homodimers. ALK2-WT (light blue) forms homodimers, while ALK2-R206H (dark blue) does not, and both are unaffected by ActA. (**B**) Effect of complex formation with ACVR2A or ACVR2B on the homomeric clustering of ALK2-R206H. The recruitment of the mainly monomeric ALK2-R206H into clusters by the type II receptor depends on the extent of homodimerization of the type II receptor. Thus, the largely dimeric ACVR2B can induce ALK2-R206H clustering already without ligand, while ActA enhances this effect due to increasing ACVR2B homomeric complex formation and its heteromeric interactions with ALK2-R206H. On the other hand, ACVR2A cannot induce clustering of ALK2-R206H, as both receptors are mainly monomeric in the absence of ligand. Upon binding of ActA, ACVR2A forms homodimers and can then induce clustering of ALK2-R206H. The homomeric clustering of ALK2-R206H leads to aberrant signaling to SMAD1/5/8 without a need for phosphorylation by the type II receptor.

## Data Availability

All data generated or analyzed during this study are included in this article and Appendix A. Additional relevant data are available from the corresponding author upon reasonable request.

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
