# Peer review of "The Activation of the Fibrodysplasia Ossificans Progressiva-Inducing ALK2-R206H Mutant Depends on the Distinct Homo-Oligomerization Patterns of ACVR2B and ACVR2A"

_cells, 2024, doi:10.3390/cells13030221_

Round 1

Reviewer 1 Report

Comments and Suggestions for Authors

The study utilized patch/FRAP analysis to explore interactions among ACVR2A, ACVR2B, wild-type ALK2, and ALK2 R206H mutation. ACVR2B formed homomeric complexes activated by ActA, whereas ACVR2A required ActA for homodimerization. Wild-type ALK2 exhibited homomeric complexes unaffected by ActA, while ACVR2B formed ActA-enhanced heterocomplexes with ALK2 R206H or wild-type ALK2. ACVR2A mainly interacted with wild-type ALK2. ACVR2B induced ActA-independent ALK2 R206H clustering, requiring ActA for enhancing wild-type ALK2 oligomerization. ACVR2B activated ALK2 R206H without ligand, while ACVR2A's activation was weaker and needed ActA, as evidenced by pSMAD1/5/8 expression and BRE Luc reporter analysis.

This study conducted a comprehensive analysis of homo- and heterodimer formation involving ACVR2A, ACVR2B, wild-type ALK2, and R206H mutation, providing insights into the distinctive homo-oligomerization patterns of ACVR2B and ACVR2A and their impact on the activation of the FOP-inducing ALK2-R206H mutant.

Comments:

In the Discussion section, there is considerable redundancy in describing the results and citing figures. The author is encouraged to delve into the novelty, extent, and hypothetical models derived from the results. Specifically, a discussion on the pathological role of the R206H mutation in Fibrodysplasia Ossificans Progressiva and the molecular regulation, with a focus on SMAD1/5/8 activation, would be valuable.

Activin, BMP, and TGFβ activate SMAD1/5/8 in different ways, potentially dependent on the interaction with ACVR2A, ACVR2B, and ALK2. To enhance the robustness of the study, it is suggested to incorporate at least one alternative ligand, such as BMPs or TGFβ, to validate SMAD1/5/8 activation.

Line 715: clarification is needed for the statement "Here, ALK2-WT yielded % homodimerization of 23%." The meaning of "% homodimerization of 23%" requires explanation.

Author Response

We thank the reviewer for the thorough review of the manuscript and the constructive remarks.

Our point-by-point responses to the comments, with a list of the changes made, follow.

(1) "In the Discussion section, there is considerable redundancy in describing the results and citing figures. The author is encouraged to delve into the novelty, extent, and hypothetical models derived from the results. Specifically, a discussion on the pathological role of the R206H mutation in Fibrodysplasia Ossificans Progressiva and the molecular regulation, with a focus on SMAD1/5/8 activation, would be valuable."

We thank the reviewer for this comment. As suggested, we reduced the redundancy and removed multiple citations of the figures (see lines 731, 740, 776, 789-791, 794, 800, 812, 827, 831, 834, 835).

Moreover, we have added a discussion on the pathological role of the R206H mutation in FOP. This appears in lines 904-910 of the revised version.

(2) "Activin, BMP, and TGFβ activate SMAD1/5/8 in different ways, potentially dependent on the interaction with ACVR2A, ACVR2B, and ALK2. To enhance the robustness of the study, it is suggested to incorporate at least one alternative ligand, such as BMPs or TGFβ, to validate SMAD1/5/8 activation:"

We agree with the reviewer that comparison with the signaling by other ligands in the same cells is beneficial. We regret that we did not emphasize clearly enough in the former version that such studies were already conducted by us in the same cell line, and published recently (Ref. 50). In the revised version, we have entered a section clarifying that signaling to SMAD1/5/8 by BMP9, and to SMAD2/3 by ActA, are reported in that publication, along with studies establishing the receptors involved (ACVR2A, ACVR2B, ALK2, ALK3 and ALK4). We have also explained that in the current manuscript we focus on aberrant signaling to SMAD1/5/8 by ActA via the ALK2-R206H mutant (see lines 864 - 873 in the revised version).

(3) "Line 715: clarification is needed for the statement "Here, ALK2-WT yielded % homodimerization of 23%." The meaning of "% homodimerization of 23%" requires explanation."

As requested, we have clarified this sentence (lines 773-774).

Reviewer 2 Report

Comments and Suggestions for Authors

This manuscript further develops the important findings the molecular mechanism of FOP reported in “Ramachandran et al., EMBO J, 2021” and “Katagiri et al., Nat Commun, 2023”. The authors findings that a specific BMP type 2 receptor, ActR-2b is involved in the activation of mutant ALK2 by FRAP measurements. This finding may provide a new perspective for elucidating the pathogenesis and developing therapeutic drugs for the diseases caused by ALK2 mutations, including FOP and DIPG. This manuscript results and discussion are well done, but I feel that some data need to be corrected.

I understand that the protein expression levels of the plasmid used in FRAP measurements has been confirmed in a previous paper. However, in this paper, in order to conclude the importance of ActR-2b, it is necessary to at least show that the protein expression levels of ActR-2a, ActR-2b, and ActR-2b(KD) are equivalent.

All vectors used in FRAP measurements are tagged with N-terminal. N-terminal region is important for ligand binding. Please show that there is no difference in phosphorylated Smad or BRE-Luc activity between N-terminal tagged vectors and tag-free vectors.

One of important finding in this paper is that ActR-2b(KD) activate ALK2(R206H). Please confirm that the ActR-2b(KD) construct used in this experiment has lost its kinase activity by checking that it is not activated by BMP stimulation (phosphorylated Smad or BRE-Luc).

In Figures 1 to 6, it is difficult to understand which receptor is being tested to bind to which receptor. I think adding illustration of receptors to understand the experiment.

Author Response

We appreciate the constructive comments by the Reviewer. We have considered and provide below point-by-point responses to all (see below).

(1) "I understand that the protein expression levels of the plasmid used in FRAP measurements has been confirmed in a previous paper. However, in this paper, in order to conclude the importance of ActR-2b, it is necessary to at least show that the protein expression levels of ActR-2a, ActR-2b, and ActR-2b(KD) are equivalent."

We have accepted this suggestion, and have added a new supplementary figure (Figure S3) where we show that the levels of all the receptors employed in the FRAP studies are equivalent (Figure S3, and lines 421-424).  

(2) "All vectors used in FRAP measurements are tagged with N-terminal. N-terminal region is important for ligand binding. Please show that there is no difference in phosphorylated Smad or BRE-Luc activity between N-terminal tagged vectors and tag-free vectors."

Following this note, we have clarified that the activity of each receptor construct utilized in the signaling experiments was verified by us in the same cell line either our recent publication (Ref. 50) or in the current study (detailed explanation appears in lines 616-621). An independent line of evidence is provided by our demonstration that the binding of the ligand (ActA) is retained by the epitope tagged receptors, as evidenced by the enhancement of both homomeric and heteromeric interactions between the epitope-tagged receptors in the presence of ActA. In view of the above, we have also explained that although some differences between tagged and untagged receptors cannot be excluded, it appears that the major signaling features are retained in the tagged receptors. We relate to these issues in the text (lines 616-627).

(3) "One of important finding in this paper is that ActR-2b(KD) activate ALK2(R206H). Please confirm that the ActR-2b(KD) construct used in this experiment has lost its kinase activity by checking that it is not activated by BMP stimulation (phosphorylated Smad or BRE-Luc). "

In response to this note, we have added a new supplementary figure (Figure S4), showing that transfection of U2OS cells with myc-ACVR2B-WT yields a significant amount of activation of the SMAD1/5/8 pathway as determined by the BRE-Luc luciferase assay, while transfection with the myc-ACVR2B-KD mutant lacks any effect. Moreover, ActA (which we preferred to use here due to the importance of ActA-mediated aberrant signaling to SMAD1/5/8 for the FOP disease) enhanced the transcriptional activation of BRE-Luc in U2OS cells expressing myc-ACVR2B-WT, but this effect was fully abrogated in cells expressing the KD mutant. These studies appear in Figure S4, and in the text (lines 607-608).

(4) "In Figures 1 to 6, it is difficult to understand which receptor is being tested to bind to which receptor. I think adding illustration of receptors to understand the experiment."

We thank the reviewer for this note, which helped us to make the manuscript clearer. In response to this suggestion, we have added a new supplementary figure (Figure S1) which presents a schematic illustration describing the various variations of the patch/FRAP experiments employed in the current studies. Each panel in this figure depicts a specific type of measurement. Furthermore, in each of the patch/FRAP experimental figures (Figures 1-6), we have added in the legend a reference to the specific panel in the schematic Figure S1, where it is shown which receptor is immobilized and which is measured by FRAP (lines 344-346; 393-395; 439-441; 459-460; 515-517; 541-542).

Round 2

Reviewer 1 Report

Comments and Suggestions for Authors  

The authors have responded to the comments with adequate explanations. I suggest the article is now suitable to be published in Cells.

Reviewer 2 Report

Comments and Suggestions for Authors

The authors accepted and kindly responded to the reviewers' comment. I think this paper worth to be published in Cells.